# Effects of Surface Modification and Heat Treatment on the Storage and Application Properties of Waterborne Zinc-Based Paint

**Liran Liu** [1,2,†], **Yangtao Zhou** [2,†], **Qingpeng Li** [1,3,*], **Yuejun Yin** [4], **Wei Zhang** [3] and **Na Wang** [1,3,*]

1   Liaoning Provincial Key Laboratory of Synthesis and Preparation of Special Functional Materials, Shenyang University of Chemical Technology, Shenyang 110142, China
2   Shenyang National Laboratory for Materials Science, Institute of Metal Research, Chinese Academy of Sciences, Shenyang 110016, China
3   Shenyang Advanced Coating Materials Industry Technology Research Institute, Shenyang 110330, China
4   Shenyang Hangda Technology Co., Ltd., Shenyang 110043, China
*   Correspondence: qpli001@163.com (Q.L.); iamwangna@sina.com (N.W.)
†   The authors contributed equally to this work.

**Abstract:** The high activity of metallic zinc particles with water, and consequently the short pot lift of a mixed waterborne organic zinc-rich paint, are the most well-known problem for their application. In this study, zinc powders were modified by silane-crosslinked potassium silicate and the paint's pot life was prolonged. Electron microscopy analysis showed that the zinc spheres in the waterborne paint were encapsulated by the shell consisted of silane-crosslinked potassium silicate and resin. The modification allowed the paint stay fluid after storage for 36 h. Nevertheless, the thickened shell was found to deteriorate the cathodic protection provided by the zinc particles. As a repair strategy, the post-heat treatment performing on the coating could awaken the protective effect of zinc powders. The anti-corrosion performance of the repaired coatings was confirmed by electrochemical tests and salt spray tests.

**Keywords:** waterborne zinc−based paint; storage stability; surface modification





## 1. Introduction

Zinc-rich coatings (ZRCs) are widely used in heavy duty industries to protect steel structures in corrosive environments, such as bridges, offshore drilling platforms, outdoor power towers and ship steel decks [1–4]. In recent years, volatile organic solvents have been strictly restricted due to the increasing environmental pollution problems and the rising awareness of environmental protection. Waterborne (WB) zinc-based paints are increasingly replacing solvent borne paints [5]. Resin primers enriched with zinc are the most widely used waterborne organic zinc-rich paints (ZRPs) in applications. The advantages of waterborne organic ZRPs include quick curing ability, good flexibility and adhesion, less risk of mud cracking, high tolerance of variable surface conditions, etc. [6,7].

For zinc-rich inorganic paints, the higher porosity, owing to a high weight of zinc particles (often 75~85%) in the coatings, usually results in the poor shielding effect and less conductivity among zinc powders [8,9]. In the past years, considerable efforts have been paid to overcome the problems. Nonmetallic pigments such as clay nanolayers, montmorillonite and $ZnO_2$ particles have been utilized to reduce the porosity, as well as the amount of zinc powders [10,11]. The addition of conductive pigments, such as grapheren, carbon nanotube, etc., are believed to be more helpful to improve the anti-corrosion performance of the coatings [12–14].

Meanwhile, different corrosion inhibitors were also used to modify zinc powder. For example, Zhu et al. modified zinc powder with rare earth lanthanum to study the electrochemical behavior of modified zinc powder and improve the corrosion resistance

of zinc-rich paints [15]. Bastos and others use cerium nitrate-modified zinc powder to improve the weldability of the coating and corrosion resistance of the primer [16]. Alipour et al. [17] functionalized molybdate by adding corrosion inhibitors, and used 3-bromopropyl trimethoxysilane (MPTMS) to improve the dispersion of mesoporous silica (MCM-41) particles. A composite coating was prepared by using modified MCM-41 particles and zinc powder as makeup filler and active molybdate base material. Through electrochemical experiments, the results showed that Mo was released by the functionalized molybdate in NaCl solution as an anode inhibitor, and the open-circuit potential of the coating increased with the extension of time in NaCl solution, and the coating had a good self-healing function. Wu et al. [18] studied the effect of RGO on the conductivity of the polyurethane coating system, and used hydrazine hydrate to reduce graphene oxide (Hummers) to obtain RGO suspension. It was then mixed with Cationic Waterborne Polyurethane (CWPU), and the CWPU/RGO composite coating was obtained through uniform ultrasonic dispersion. The experimental results show that the coating conductivity of 0.28 S·m$^{-1}$ is nearly 10 orders of magnitude higher than that of CWPU ($5.66 \times 10^{-11}$ S·m$^{-1}$) when the addition of RGO is 3.2 wt%. The results showed that the addition of reduced GO significantly improved the conductivity of the coating. Meanwhile, Meng et al. [19] studied the effect of Graphene Platelets (GP) on the electrical conductivity of EP coatings. They first dissolved the graphene sheet in azomethyl pyrrolidone, then added surfactant (J2000) to modify the graphene sheet (J2000-GP), removed too much J2000, dissolved in bisphenol A diglycrin ether (DGEBA) and further modified the modified graphene sheet (m-GP). Finally, this was mixed with EP to obtain m-GP/EP composite coating. The experimental results show that the resistivity of the composite coating decreases from $10^{14}$ Ω·cm to $10^{5.3}$ Ω·cm when the additive amount of GP is 2.3 vol %. Therefore, the addition of graphene sheet can improve the electrical conductivity of the coating, and at the same time, the mechanical properties, surface roughness and thermal stability of the coating are also improved.

An important engineering problem with WB organic ZRPs, which is relatively less studied, is the poor stability and short pot life after mixing the multi-components. Incorporating such a high weight of zinc powder into a WB resin agent system faces a significant challenge in that zinc is highly reactive with water during the application process [20–22]. Consequently, most WB paints have a visible end of pot life. The viscosity of the paint rises dramatically within two hours in general. One unique technique is to modify zinc particles with water-free curing agents. It was reported in a patent that the zinc dust was dispersed into a silane hydrolyzate, and prevented from reacting with water [23]. However, detailed research results were not available to the authors' knowledge. Regarding some other WB paints, the problem is different but more pronounced that they do not have a visible end of pot life. The mixed paint stays fluid, but the gloss and performance are significantly degraded. For those paints, the application must be controlled in a time window which largely depends on the operation conditions and the painters' experience. Hence, it should be valuable to have an approach that can reduce the application risk of the paint beyond the useful pot life.

Herein, the surface modification of zinc powders by silane cross-linked inorganic potassium silicate was utilized to improve the shielding effect and storage stability of a WB acrylic ZRC. The prepared zinc powders and coatings were systematically characterized. On the other hand, we proposed a post heating strategy to remedy the property degradation of the mixed zinc-containing resin paint after long-term storage.

## 2. Experimental

### 2.1. Materials

Industrial grade SE200 acrylic resin was purchased from Shenyang Zhongke Anti-corrosion Engineering Technology Center and used as a binder in the paint. SE200 is a waterborne acrylic resin emulsion with an effective solids content of $45 \pm 2\%$, a milky white appearance and an overall viscosity (cps/25 °C) of 200–5000. It is a dense film with a good balance of hardness and flexibility, and is a fast-drying waterborne emulsion.

Industrial grade spherical zinc powders with particle sizes of 0.1–10 μm were purchased from Hunan New Welling Company. (Changsha, China). High modulus industrial grade potassium silicate, with a modulus of 4~6 M, was purchased from Wuhan Hyster Technology Co., Ltd. (Wuhan, China) Silane coupling agent A−187 was purchased from Nanjing Nengde New Material Technology Co., Ltd. (Nanjing, China) Alkylphenol polyoxyethylene ether (OP−10) and dispersant were of industrial grade and purchased from Tianjin Bodi Chemical Co., Ltd. (Tianjin, China) and German BYK Chemical Company(Munich, Germany), respectively.

### 2.2. Preparation of Waterborne Zinc-Based Paints

Waterborne organic acrylic resin was modified by cross-linking high modulus potassium silicate inorganic material and silane coupling agent A-187. Firstly, 0.5 kg of inorganic high-modulus potassium silicate (4–6 M) was stirred at a speed of 1500 rpm at room temperature (25 ± 5 °C) for 0.5–1 h. Then, 0.25 kg of silane coupling agent A-187 was added in sequence. The stirring speed was adjusted to 2000 rpm, and the mixing temperature was adjusted to 50 ± 2 °C. Then, 3 kg of waterborne organic acrylic resin was added and shaken at 3000 rpm for 4–5 min at 50 ± 2 °C. Finally, the modified resin was filtered through a 500-mesh filter to obtain a modified shell adhesive with good dispersibility. The chemical process is schematically illustrated in Figure 1a. High-modulus potassium silicate is firstly cross-linked with the hydrocarbon bond of silane coupling agent, and then cross-linked with aqueous acrylic resin.

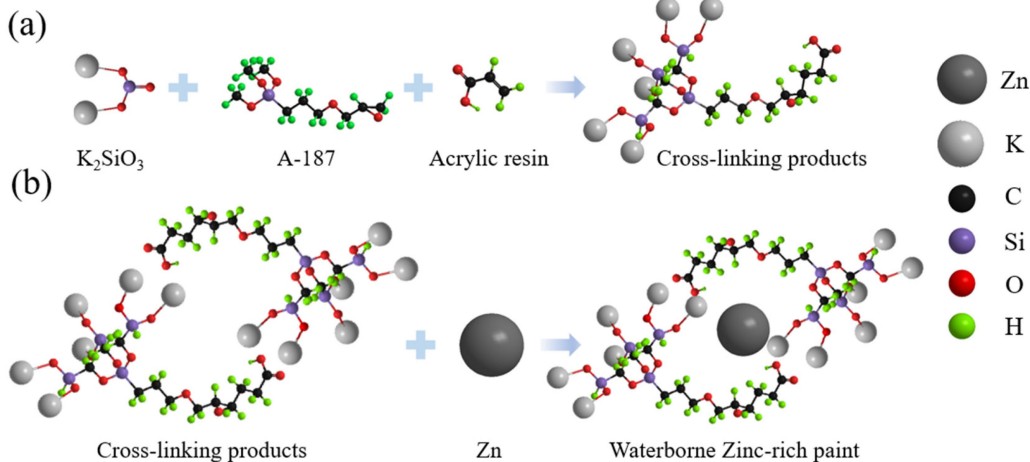

**Figure 1.** Schematic diagram of the surface modification on a zinc powder: (**a**) Schematic diagram of the silane modification principle (**b**) Schematic diagram of the zinc powder coating principle.

Preparation of waterborne zinc-based paint: 3 kg of modified resin was stirred at room temperature (25 ± 5 °C) with a high-speed mixer at a speed of 1500 rpm for 10 min. 0.1 kg of OP-10 and 0.1 kg of BYK dispersant were added, and the mixture was stirred at room temperature for 1 h. Finally, 10 kg of spherical zinc powder (0.1–10 μm) was added to the stirred solution, and stirred at 2000 rpm for 2 h to prepare a WB zinc-based paint. As a result, the zinc spheres were modified as shown in Figure 1b, forming a resin shell.

Q235 low-carbon steel sheets (150 mm × 70 mm × 1 mm) were employed to evaluate the anticorrosion performance of zinc-pigmented coatings. Sandblasting was performed on the steel surface to ensure that the paint film could be firmly attached to the steel base. Q235 ordinary carbon structural steel is also known as plain plate. The yield value is about 235 MPa. Due to carbon content (C 0.14~0.22%, Mn 0.30~0.65, Si ≤ 0.30, S ≤ 0.050, $p \leq 0.045$) moderate, good comprehensive performance, strength, plasticity and welding properties are a better match, as well as the most widely used. Therefore, it is selected as the substrate for coating protection in this paper. In this study,

the same paints were prepared using the same formulation, then stored for different times, and then applied onto the steel substrate using an air sprayer with a pressure of 0.01–0.05 MPa and a spraying distance of 150–200 mm. The thickness of the coating was $100 \pm 10$ μm.

### 2.3. Characterization Methods

Fourier infrared spectroscopy tests were carried out using Nicolet 560 infrared spectrometer. In the test, potassium bromide was used as the medium, and the sample was mixed with potassium bromide according to the mass ratio, and then pressed into tablets after grinding. Scanning range was chosen between 400–4000 cm$^{-1}$.

In order to observe the morphology of zinc powder and waterborne zinc-based paint, the prepared paint was diluted with distilled water, and the supernatant was taken and dropped on a copper grid. An FEI Tecnai F20 transmission electron microscope was used. The surface morphology was observed using an FEI Inspect F50. The phase analysis of the coatings was carried out using the XPERT-PRO test system with Cu(Ka) as the diffraction source and an angular resolution of 0.02°. Phase analysis was performed using software Jade version 6.5 and database JOPDS (ICDD).

Corrosion resistance of the coatings was evaluated by neutral salt spray tests using HDYW-120 salt spray box. Specimens were exposed in a standard salt spray box according to the standard ASTM B 117~09. The treated samples are placed in the salt spray tester. Each sample is placed on one side at an angle of 45° to the vertical, with the samples spaced apart to avoid interfering with each other and affecting the test results. After the salt spray exposure test, the samples are carefully removed, rinsed with distilled water to remove any remaining impurity particles on the surface and then dried at room temperature.

In order to verify the electrochemical properties of the coating, the open-circuit potential and dynamic potential polarization curves were tested using a CHI6600E electrochemical workstation. A plastic tube with an inner diameter of 4 cm is fastened to the surface of the sample with nuts and bolts. This assembled electrochemical cell was then injected with a 5 wt% sodium chloride solution as a corrosive electrolyte. A standard three-electrode system was used, with a standard calomel electrode (SCE) as a reference electrode and platinum as a counter electrode, and the prepared sample as the working electrode. The polarization curve was scanned at a speed of 5 mV/min, the polarization data was collected by the computer, and then the curve was fitted with C-View software for data analysis. The open-circuit potential sweep time is 180S.

### 3. Results and Discussion

#### 3.1. Waterborne Zinc-Based Paint

The silane coupling agent A-187, high modulus potassium silicate (HMPS) and graft cross-linked modified products, and the final aqueous film-former were characterized by infrared spectroscopy, Figure 2. The modification cross-linking effect was verified by the vibration fingerprints of the molecules. The stretching vibration absorption bands of 2960 cm$^{-1}$ and 2840 cm$^{-1}$ of -CH$_3$ belonging to silane coupling agent A-187 appeared in the spectrum of the modified product of A-187 modified potassium silicate and the spectrum of the final film-forming agent. The Si=O characteristic peak 1635 cm$^{-1}$ and the Si-O characteristic peak 1070 cm$^{-1}$ of high modulus potassium silicate also appeared [24]. However, compared with the characteristic peaks of pure coupling agent A-187 and high-modulus potassium silicate, the intensity of some of these characteristic peaks of the cross-linked products was significantly weaker, indicating that the silane coupling agent successfully cross-linked the modified potassium silicate and the aqueous acrylic resin.

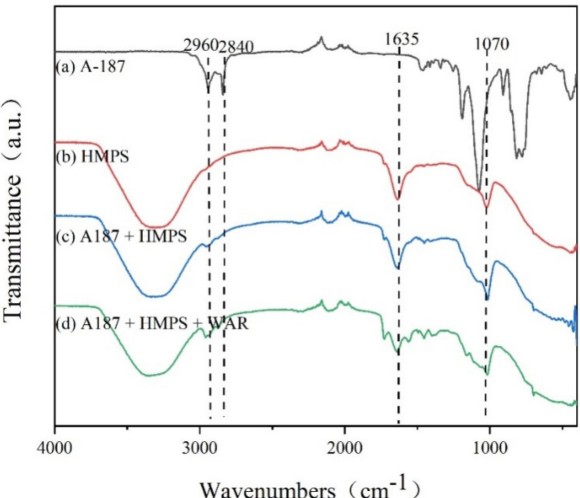

**Figure 2.** FTIR spectra of (**a**) A−187, (**b**) High modulus potassium silicate (HMPS). (**c**) A−187 + HMPS and (**d**) A−187 + HMPS + Waterborne acrylic resins (WAR).

The morphology of zinc spheres could be visualized by electron microscopy. Figure 3a is the SEM image of pristine zinc spheres. The particle sizes are ranged from 0.1 μm to 10 μm. TEM image of an individual zinc particle is given in Figure 3b. The particle surfaces are clean and smooth as seen in both the SEM and TEM images. By contrast, the modified zinc powders were encapsulated by apparent shells on the surfaces. Figure 3c is the SEM image of a coating in which the zinc particles are covered by the uniform binder film, and the fuzzy structure circled by the red circle is the shell structure of the zinc powder surface. The prepared paint was ultrasonically dispersed with water, and then the supernatant drops were taken on a copper grid for TEM characterization. The surface shell structure of the zinc spheres can be visualized by transmission electron microscope characterization, as shown by the red arrow in Figure 3d. The distinct shell layer can be measured to be about 100 nm in thickness. The images clearly indicate that the additions of silane and inorganic potassium silicate form a dense shell upon a zinc particle, and protect it from the reaction with water.

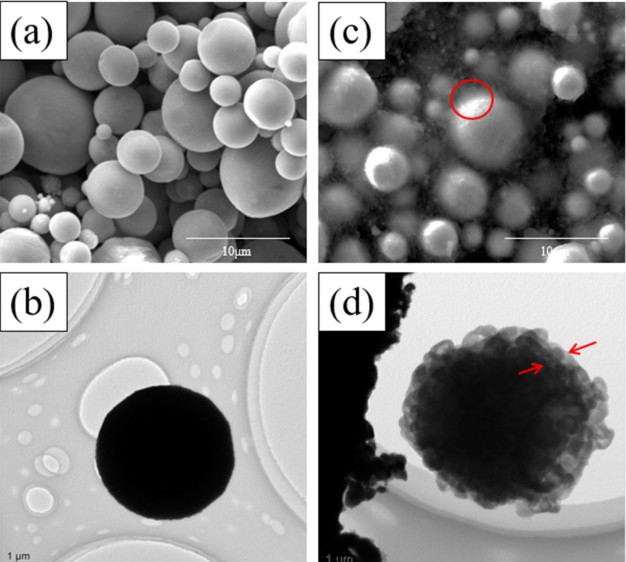

**Figure 3.** (**a**) SEM image of pristine zinc powders. (**b**) TEM image of the single zinc particle with smooth surface. (**c**) SEM image of the WB zinc-based coating. (**d**) TEM image of an individual zinc particle encapsulated by a shell consisting of silane-crosslinked potassium silicate and resin.

### 3.2. Storage Stability

Because of the reaction activity between the metallic zinc powder and $O_2$, $H_2O$ in the WB system [15], the viscosity of the prepared paint tends to increase, leading to difficulty in application. In general, the pot life of a two-component acrylic zinc-rich paint is often less than 2 h. From the data in Table 1, however, we can see that the silane modification obviously increased the stability of the paint. After being stored at room temperature for 36 h, the paint did not change much in the outflow time and the viscosity.

**Table 1.** Outflow times of WB zinc-based paints after storage for different times.

| Storage Time/h | Outflow Time/s | State |
|---|---|---|
| 0 | 25 | Normal |
| 6 | 25 | Normal |
| 16 | 25 | Normal |
| 20 | 26 | Normal |
| 36 | 30 | Normal |

If the zinc powder is not covered with a modified resin in a one-component waterborne zinc-rich paint, the strong electrical conductivity of the zinc powder can lead to short term hydrogen precipitation reactions, which can cause the paint to fail. The purpose of the modified resin coating is to reduce the electrical conductivity, thus avoiding hydrogen precipitation reactions and increasing storage stability. However, if the shell layer is too thick, the zinc powder will always be in contact with the zinc powder, making it difficult to sacrifice the anode for cathodic protection.

To investigate the component change of the paint during storage, XRD and SEM tests were performed on the samples. As shown in Figure 4a, both the as-prepared and the stored coatings are mainly composed of metallic Zn. The diffraction peaks of resin and potassium silicate could not be easily distinguished due to the poor crystallization nature. Using the paint stored for 36 h, we spayed the coating on a steel plate and examined it under an electron microscope. As displayed in Figure 5, we could find that the surface shells on zinc particles are thickened. The XRD and SEM results indicated that the modification effectively prevented the contact between zinc and water by forming a shell structure which became thicker along with the storage periods.

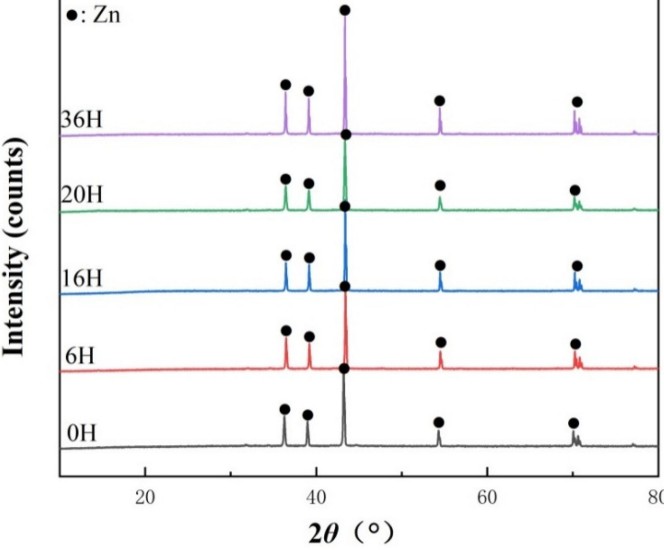

**Figure 4.** XRD patterns of WB zinc-based coatings prepared by using the paints stored for different periods, i.e., 0H, 6H, 16H, 20H, 36H.

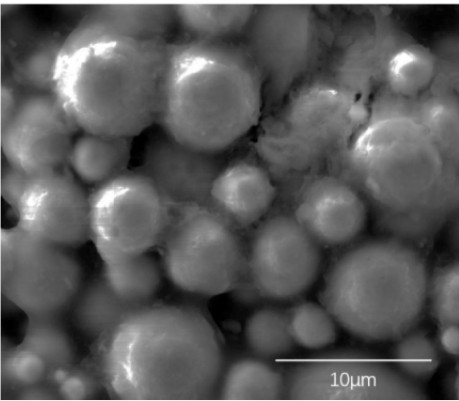

**Figure 5.** SEM image of WB zinc-based coating after 36 h storage.

The electrochemical performance of the coatings prepared using the stored paint was compared by the change of open circuit potentials (OCPs) and the polarization tests. Figure 6 is the OCP change measured on the coatings immersed in a 5 wt% NaCl solution for 1000 h. The overall open circuit potential change trend for the sample 6H was the same as that of the coating 0H. After being immersed for more than 400 h, the cathodic protection by zinc powders was significantly reduced, and the open-circuit potentials slowly rose to the OCP of steel substrate [25,26]. The figure also shows that long-term storage of the paint could also hurt the anti-corrosion performance. For the sample 16H, the cathodic protection of zinc was almost lost.

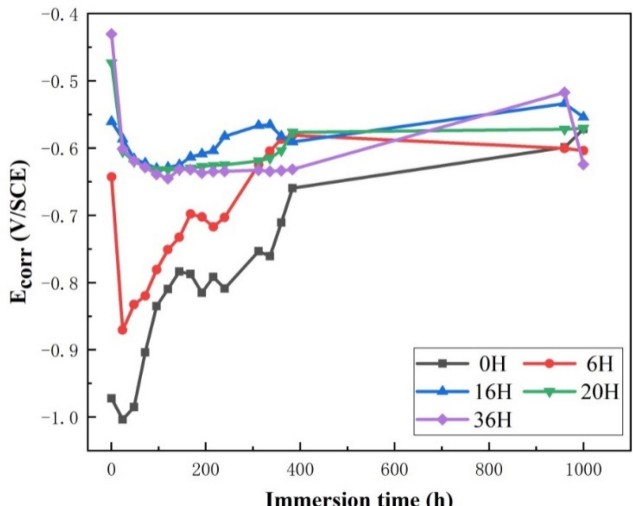

**Figure 6.** Open circuit potential of WB zinc−based coatings after storage for different times (0H, 6H, 16H, 20H, 36H).

The corrosion resistance of ZRCs could be also evaluated by the neutral salt spray method [27]. The as-prepared paint and the stored paint were applied on steel plates and exposed to the corrosive environment over 1000 h. Figure 7 is the optical surface image of the coated steel plates. All the samples were covered with white rust after 1000 h of salt spray testing. No red rust was observed on the coatings 0H and 6H, indicating that the ZRCs provided effective protection to the substrate. On the surfaces of the coatings 16H, 20H and 36H, however, the emergence of red steel corrosion indicated the decrease of anti-corrosion performance of the paints stored for long period. On the basis of the above analysis, we could conclude that the thickening of the shell layers upon the zinc spheres leads to the failure of cathodic protection.

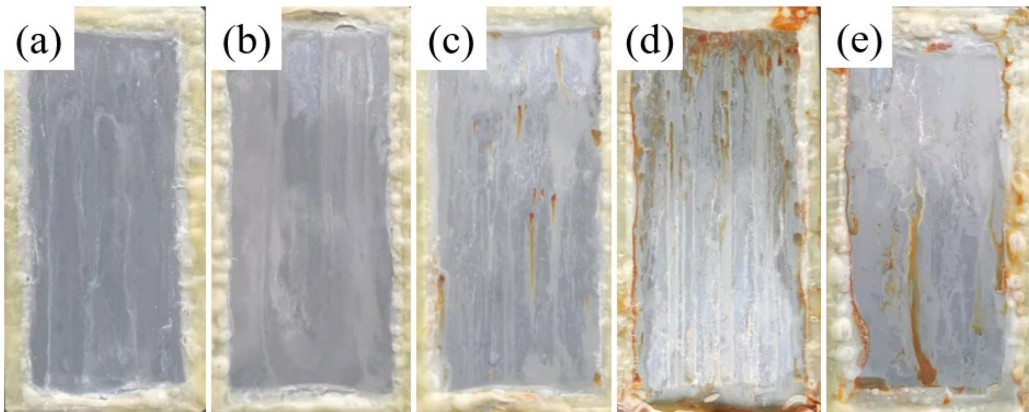

**Figure 7.** The optical images of WB zinc-based coatings after corrosion test at 1000 h in 5 wt.% NaCl: (**a**): 0H, (**b**): 6H, (**c**): 16H, (**d**): 20H and (**e**): 36H.

### 3.3. Effect of Post-Heating Treatment on the Coatings

As mentioned above, the WB zinc-based paint after 36 h storage remained excellent paint flow, but its anti-corrosion performance was substantially degraded. Here, a post-heating strategy was proposed to repair the long-term storage-induced coating degradation.

XRD results indicated that the baking generated a small amount of ZnO (Figure 8). Baked at 400 °C and 500 °C, the coating did not differ significantly, with the main actor consisting of Zn and a small amount of ZnO. In contrast, when the baking temperature was increased to 600 °C, the ZnO content of the coating increased significantly.

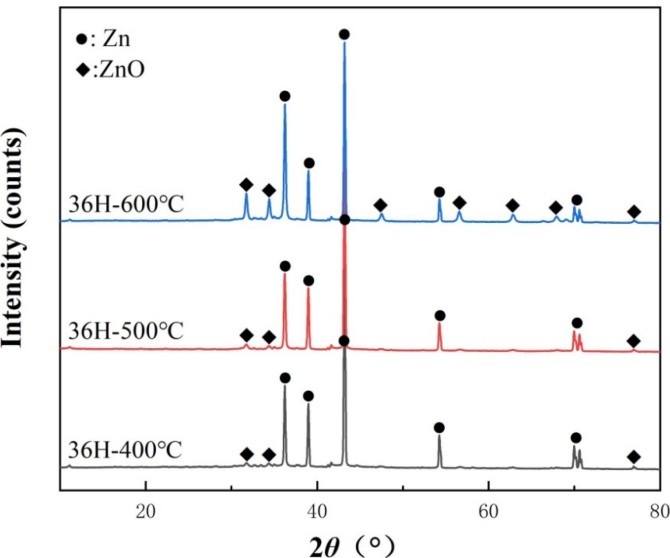

**Figure 8.** XRD patterns of the stored WB zinc-based coatings treated at different temperatures.

The three samples were further characterized by an electron microscope. It can be seen from Figure 9a,b that the resin binder shrinks due to the high temperature treatment. The morphology of zinc powders in the coating did not change significantly after baking at 400 °C and 500 °C. The zinc particles are still surrounded by potassium silicate layers. Nevertheless, according to the XRD results, it could be conceived that the high temperature baking caused a large number of cracks to appear in the inorganic surface layers, allowing the Zn particles to react with oxygen and generate ZnO. When the baking temperature was increased to 600 °C, a large number of spikes appeared on the surface of the zinc spheres.

This was probably due to the breakdown of the shell layer, and caused the rapid oxidation of zinc; it was also confirmed by the large amount of ZnO detected by XRD.

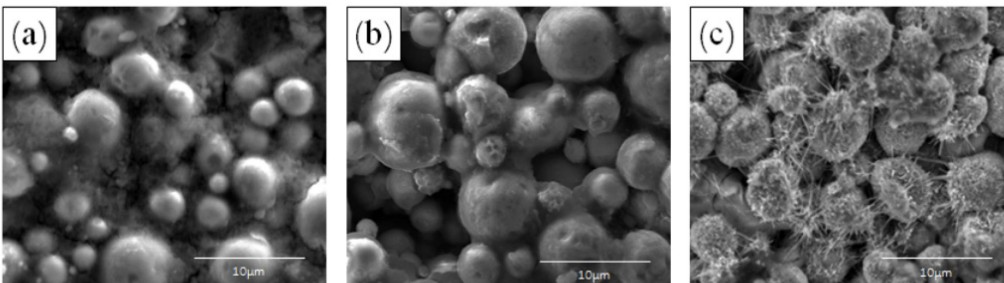

**Figure 9.** SEM images of WB zinc-based coatings baked at different temperatures. (**a**) 36H-400 °C, (**b**) 36H-500 °C and (**c**) 36H-600 °C.

According to the EDS map of zinc, we could find that the porosity between zinc spheres is filled by ZnO. This confirms the SEM observation that the heat treatment broke the surface shell and exposed the fresh zinc particles. At the same time, the resin was carbonized by the treatment. As shown in Figure 10d–f, the carbonization results in a more even distribution of the carbon, and probably the enhancement of the electrical conductivity in the coating [14].

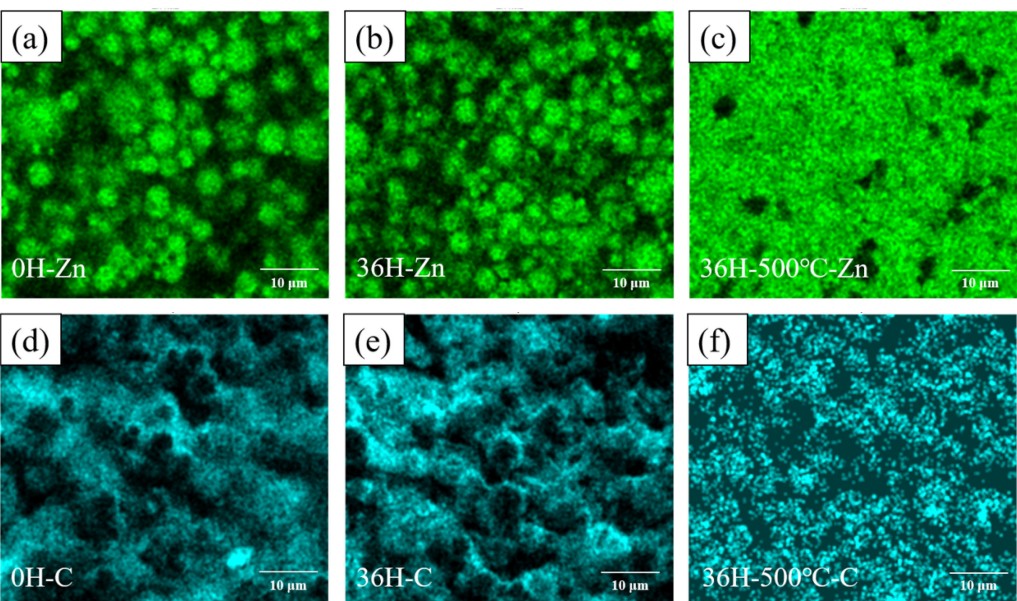

**Figure 10.** EDS zinc maps of the coatings (**a**): 0H, (**b**): 36H and (**c**): 36H-500 °C. EDS carbon maps of the coatings. (**d**): 0H, (**e**): 36H and (**f**): 36H-500 °C.

Figure 11 shows the OCP variations of the coatings treated at different temperatures. It seems that the treatment at 400 °C did not improve the cathodic protection of the coating. After baking at 500 °C, the coating possessed the lowest OCP at the early 250 h immersion in a 5 wt% NaCl solution, indicating that a large part of the fresh zinc particles have been exposed at this temperature. The OCP variation of the coating treated at 600 °C follows the same trend as that of the sample 36H-500 °C.

Figure 12 shows the potentiodynamic polarization curves of WB zinc-based coatings after baking at 400–600 °C in a 5 wt% NaCl solution. The fitted results based on the polarization curves are shown in Table 2. It can be seen that after baking at 400 °C, 500 °C and 600 °C, the corrosion potentials of the samples do not differ significantly. The corrosion current of the samples 36H-500 °C is the largest and the polarization resistance is the

smallest, which means that the zinc particles provided an effective cathodic protection to the steel substrate.

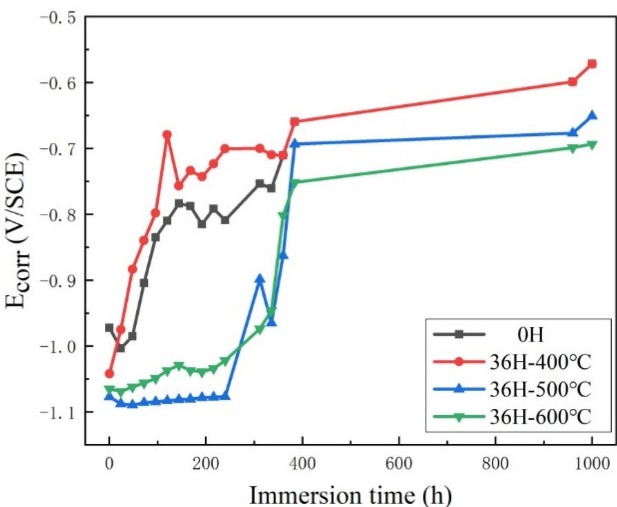

**Figure 11.** Open−circuit potential diagrams of the as-prepared WB zinc−based coating (0H) and the coatings baked at temperatures 400 °C, 500 °C and 600 °C.

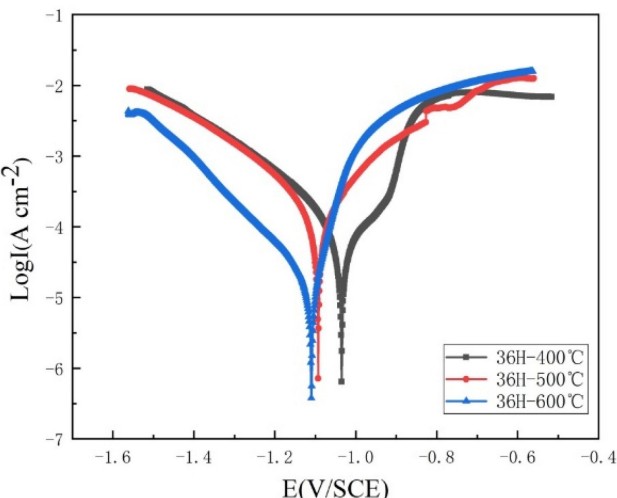

**Figure 12.** PDP curves of WB zinc−based coatings baked at different temperatures: 36H−400 °C, 36H−500 °C and 36H−600 °C.

**Table 2.** The electrochemical parameters of WB zinc-based coatings baked at temperature after storage for 36 h calculated from Tafel plots.

|  | Icorr/A cm$^{-2}$ | Corrosion Potential (V) | Rp/$\Omega$ cm$^2$ |
| --- | --- | --- | --- |
| 36H-400 °C | $7.93 \times 10^{-5}$ | −1.04 | 420.1 |
| 36H-500 °C | $1.45 \times 10^{-4}$ | −1.09 | 227.7 |
| 36H-600 °C | $1.92 \times 10^{-5}$ | −1.11 | 1113.0 |

The performance of the heat-treated coating samples were also tested by a neutral salt spray test. Notably, after only 48 h, large cracks appeared on the 36H-600 °C sample (Figure 13c), leading to severe corrosion on the internal steel substrate. This suggests that carbonization of the resin bonder and severe oxidation of zinc particles during the treatment at high temperature results in the significant brittleness of the coating. In comparison, the coating sample 36H-400 °C and 36H-500 °C could effectively provide the salt spray

resistance for more than 1000 h. The white zinc corrosion products on the coating surface suggest the sacrifice dissolution of zinc particles which were exposed due to the post-heat treatment. In Figure 13a, a small amount of red rust is observed on the coating surface. Combining the OCP and potentiodynamic polarization tests, heat treatment at 500 °C should be an optimized option to repair the long-term stored WB paint.

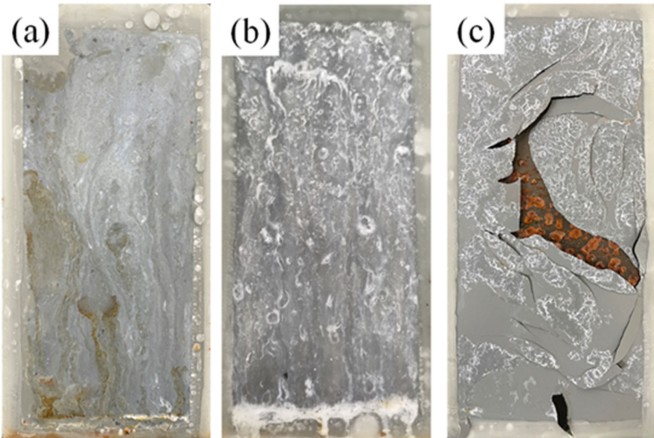

**Figure 13.** The optical images of WB zinc-based coatings (**a**) 36H-400°C and (**b**) 36H-500 °C after corrosion test for 1000 h. (**c**) The coating 36H-600 °C was severely attacked after only a 48 h salt spray test.

## 4. Conclusions

In the present work, the zinc particles were modified by Silane Coupling Agent A-187 crosslinking with inorganic high modulus potassium silicate, and used to prepare WB zinc-based paints. A shell consisting of silane and potassium silicate was observed on the zinc particle surface, which prevented its oxidation when it was mixed in an aqueous environment. As a consequence, the pot life of the mixed paint was extended. The long-term storage generated a thick shell on zinc particles and deteriorated the anti-corrosion performance, although the paint stayed fluid. On the other hand, to overcome the performance degradation of the longstanding paint, we proposed a post-heat treatment strategy performing on the coating. The treatment at 500 °C allowed the zinc particles in the coating to be re-exposed, and play a protective role.

**Author Contributions:** Methodology, L.L., Q.L. and Y.Z.; validation, L.L., Q.L. and Y.Z.; investigation, L.L., Y.Z., Q.L., Y.Y., W.Z. and N.W.; writing—original draft preparation, L.L.; writing—review and editing, L.L., Q.L. and Y.Z.; supervision, N.W.; project administration, Q.L. and N.W. All authors have read and agreed to the published version of the manuscript.

**Funding:** This work was financially supported by the Funded by Liaoning Revitalization Talents Program (XLYC2005002); the 2021 Scientific Research Foundation of Education Department of Liaoning Province (LJK Z0470); 2021 Liaoning Province, "Unveiling the List Hanging" science and tech-nology research projects (2021JH1/10400091); and Shenyang Science and Technology Plan—Major Key Core Technology Tackling Special Project (20−202−1−15).

**Institutional Review Board Statement:** Not applicable.

**Informed Consent Statement:** Not applicable.

**Data Availability Statement:** The data is available on reasonable request from the corresponding author.

**Acknowledgments:** Y.Z. acknowledges the support of SYNL.

**Conflicts of Interest:** The authors declare no conflict of interest.

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
