# Peer review of "Effects of Surface Modification and Heat Treatment on the Storage and Application Properties of Waterborne Zinc-Based Paint"

_coatings, doi:10.3390/coatings13030652_

Round 1

Reviewer 1 Report

The manuscript is well done and is suitable for publishing in Coatings after minor revisions according to the text presented below:

1. Page 3, line 91 - The authors need to present the composition of the low-carbon steel used.

2. Page 3, lines 111-115 - In my opinion this information could be avoided since it describes the requirements of the standard ASTM B 117-09.

3. How is the isolation of the samples treated in the salt spray chamber realized?

4. Page 3, line 126 - What is the reason to use such a speed; in the most cases the speed is 1 mV/sec.

5. Page 4, line 128 - The time period needed for stabilizing the open-circuit potential in that medium should be greater than 180 sec. What is the reason?

6. Figure 3 - Please, mark the shell on the image.
